# Probabilistic Risk Assessment of Dietary Exposure to Chloramphenicol in Guangzhou, China

**DOI:** 10.3390/ijerph18168805

**Published:** 2021-08-20

**Authors:** Yanyan Wang, Weiwei Zhang, Florence Mhungu, Yuhua Zhang, Yufei Liu, Yan Li, Xiaoyan Luo, Xinhong Pan, Jie Huang, Xianwu Zhong, Shaofang Song, Hailin Li, Yungang Liu, Kuncai Chen

**Affiliations:** 1Guangzhou Center for Disease Control and Prevention, Guangzhou 510440, China; wangyy13213845@163.com (Y.W.); gzcdczhangww@foxmail.com (W.Z.); pisceszyh@126.com (Y.Z.); gzliuyufei@hotmail.com (Y.L.); gzcdcliy@foxmail.com (Y.L.); gzcdclxy@hotmail.com (X.L.); panxh1987@163.com (X.P.); huangjie1026@126.com (J.H.); zhongxw.gzcdc@foxmail.com (X.Z.); anchure@163.com (S.S.); lihailin310@163.com (H.L.); 2Institute of Public Health, Guangzhou Medical University & Guangzhou Center for Disease Control and Prevention, Guangzhou 510440, China; 3Department of Toxicology, School of Public Health, Southern Medical University, Guangzhou 510515, China; mhunguf1@gmail.com

**Keywords:** chloramphenicol, residue, dietary exposure, risk assessment, @RISK

## Abstract

Chloramphenicol has been used in veterinary medicine, where its residues can remain in food of animal origin, thus potentially causing adverse health effects. This facilitated the ban for its use in food-producing animals globally, but its residues have remained ubiquitous. In this study, food commodities possibly contaminated with chloramphenicol, including livestock meat, poultry, edible viscera, fish, shrimp and crab, molluscs, milk, and eggs, were collected from domestic retail shops in all the 11 districts of Guangzhou and tested for its residue. Probabilistic risk assessment model calculations for its dietary exposure, and the margin of exposure (displayed as mean values and 5th percentile to 95th percentile ranges) were performed by using @RISK software based on a Monte Carlo simulation with 10,000 iterations. The results indicated the detection of chloramphenicol in 248 out of 1454 samples (17.1%), which averaged to a level of 29.1 μg/kg. The highest average value was observed in molluscs (148.2 μg/kg, with the top value as 8196 μg/kg); meanwhile, based on the dietary structure of a typical Cantonese, pond fish, pork, and poultry meat contributed most (about 80%) to the residents’ dietary exposure to chloramphenicol. The margin of exposure for dietary chloramphenicol exposure in Guangzhou residents was 2489, which was apparently below 5000 (the borderline for judging a health risk), particularly low in preschool children (2094, suggesting an increased risk). In conclusion, the study suggests that chloramphenicol exposure in Guangzhou residents is considerable, and its relevant health hazard, especially for preschool children, is worthy of further investigation.

## 1. Introduction

Chloramphenicol (CAP), a broad-spectrum antibacterial compound, was once (more than 20 years back) legally used to maintain animal health, both therapeutically and preventively, owing to its effects against bactericidal infection [1]. Since it was convincingly observed that CAP can induce aplastic anaemia in humans as well as some undesirable reproductive/hepatotoxic effects in animals, CAP has been banned from use in medicine, food, animal husbandry, and aquaculture production at least for 20 years in various countries and regions [2,3,4]. However, its residues remain widely detectable in various food commodities, such as eggs, meat, milk, and honey, probably due to continuous, illegal use of CAP in domestic animals [3,5]. When CAP is administered in livestock production, residues may also contaminate surface soils and remain biologically active [6,7]. Through surface run off, CAP may accumulate, thus a life-cycle between the environment, aqua-lives, and livestock for CAP may be formed, which can pose substantial risk of exposing humans by ingestion [3,8].

Notably, due to the emergence of bacterial resistance to, and the toxicity clinically observed for, CAP, concerns about its use were expressed through several regulations set by the European Community, United States, Canada, Australia, Japan, China, and Brazil [9]. The detection of shrimp samples containing CAP residue in 2001 elicited the banishment of CAP use in the food chain in Europe [9,10]. In Brazil, CAP was banned for use in food-producing animals in 2003, and a minimum risk level (MRL) of 0.3 mg/kg, the same as adopted in Europe by the European Commission, was assumed for CAP monitoring [10]. The Export Inspection Council (EIC) of India also marked the MRL for CAP at 0.3 mg/kg. Likewise, the European Food Safety Authority (EFSA) indicates that exposure to food contaminated with CAP at or below 0.3 mg/kg is unlikely a health concern with respect to aplastic anaemia or reproductive/hepatotoxic effects, but only above this limit it may become possible [5]. Factors including the level, frequency, and duration of an exposure, baseline health status, behavioral aspects, and genetics may induce and/or aggravate the ultimate adverse effects in an individual [3,11].

Moreover, the Food and Agriculture Organisation of the United Nations (FAO), under the World Health Organization (WHO), and the Veterinary World Association have expressed concerns that there is no safe level for residues of CAP or its metabolites in food, which represents an acceptable risk to consumers [12]. For this reason, competent authorities should prevent residues of CAP in food, simply by not using CAP in food-producing animals [13].

Due to its ubiquitous nature, many reports have indicated the frequent occurrence of residues of CAP as an organic pollutant in several relevant food products in China and globally [5,14]. Because China is one of the largest producers of milk, dairy, and aquatic commodities across the globe, the quality and wellbeing of its products are of supreme concern.

As early as in 2002, the Ministry of Agriculture of the People’s Republic of China issued Announcement No. 235, “Maximum Residue Limits of Veterinary Drugs in Foods of Animal Origin”, which clearly stipulates that CAP is prohibited from use and must not be detected in foods of animal origin. However, in recent years, CAP has often been detected in animal food from various regions of China. For example, in Guangdong Province CAP was detected in 17 of 150 aquatic product samples, with a detection rate of 11.3%, with a relatively high detection rate (36%) in molluscs (a marine animal) and a low rate (5.6%) [15]; another study conducted in Guangdong Province indicates the detection rates of CAP in fish, shell fish, and shrimp as 14.3, 100, and 66.7%, respectively [16]. Likewise, in Guangxi Province, which neighbors Guangdong, CAP residue was detected in 36% of shellfish samples [17]. Moreover, CAP was also detected in 28.1% of the honey samples in an inner province, Yunnan Province (27/96) [18]. These reports all indicate that the contamination of CAP residues in animal foods in China is serious, which deserves attention from the regulatory authorities.

With regards to this, the Guangzhou Centre for Disease Control and Prevention embarked on a four-year program of CAP contamination in various food commodities in Guangzhou city, from 2016 to 2019. The aim of this program was to understand the occurrence and levels of CAP in a range of food products generally consumed by citizens in all the 11 districts of Guangzhou. After obtaining the CAP contents, we conducted dietary exposure risk assessments by analyzing the margin of exposure (MOE) of CAP [19,20,21], with exposure estimations covering citizens in different age groups.

## 2. Materials and Methods

### 2.1. Sampling

In the period between January 2016 and December 2019, based on the characteristics of antibiotics used in breeding and adaptability to laboratory conditions [22], food commodities supposedly contaminated by CAP were collected from domestic retail shops. They included livestock meat, poultry, edible viscera, fish, shrimp and crab, mollusks, milk, and eggs, and they were collected from each of the eleven districts of Guangzhou [23].

Sampling was randomized to include most individual streets, based on street information acquired from local governmental authorities. Three streets (two central and one remote) were randomly selected according to each district/type of street (central or remote) with the aid of computer-generated random digits, where 33 streets (22 central and 11 remote) were encompassed as food sampling locations. Selected food commodities were obtained from supermarkets, agricultural markets, retail shops, restaurants, and family workshops by skilled investigators. In total, the study was inclusive of 1454 nominal-species food samples.

### 2.2. Analysis of CAP by Liquid Chromatography–Mass Spectrometry (LC-MS)

The process of determination of CAP residues in food was in accordance to a previously reported method using LC-MS with some modifications [24,25,26,27,28].

Sample preparation: Representative parts were taken out from the original samples. For livestock and poultry meat, aquatic products, and viscera, 500 g edible parts of each sample was mashed and mixed evenly using a high-speed tissue homogenizer. For egg, 500 g edible parts was pulped. The homogenate of each sample was divided into two parts, one for immediate determination, the other being stored at −20 °C.

Extraction: 5 g of each sample was weighed, to which 100 µL of 20 ng/mL CAP deuterated internal standard (CAP-D_5_) working solution, 5 mL of 4% sodium chloride solution, and 5 mL of acetonitrile were added. Each homogenized tissue sample was subject to ultrasonication for 5 min. Then, 10 mL ethyl acetate was added, followed by vortex extracting for 5 min and centrifugation for 2 min, at 10,000 rpm and room temperature. The supernatant (organic phase) was stored at room temperature shortly before purification.

Purification: An SPE column (MCS type) was activated with 3 mL methanol and 3 mL ethyl acetate successively, then 5 mL supernatant (organic phase) was transferred onto the MCS column. Eluant 1 was collected in a 10 mL test tube. After the supernatant passed through the column, a further elution was done by adding 2 mL ethyl acetate and 1 mL methanol. Eluant 2 was collected and mixed with eluant 1, which was named eluant 3. Eluant 3 was blow-dried with nitrogen at 50 °C, and 3 mL 4% sodium chloride solution was added to the residue for ultrasonic dissolution. Afterwards, 3 mL n-hexane was added for eddy extraction for 2 min, then centrifuged at 10,000 rpm for 2 min. The n-hexane phase was discarded, and extraction by n-hexane was repeated. The superlayer ethyl acetate phase was sucked out, blow-dried with nitrogen at 50 °C, and the residue was added with 0.1 mL methanol and dissolved by ultrasonication. Next, it was added and mixed with 0.9 mL of 10% methanol (in water), which was subjected to ultrasonication for 1 min. Finally, the supernatant was collected after centrifugation at 10,000 rpm and passed through a 0.22 μm filter membrane to the sample bottle, which was then analyzed by LC-MS.

LC-MS conditions: For LC, a C18 column (Waters BEH, 5 cm × 2.1 mm, with particles sized 1.7 μm) was used with mobile phase A (0.05% ammonia) and mobile phase B (acetonitrile). The procedure for gradient elution is shown in Table 1; the flow rate was 0.40 mL/min, analysis time 6 min, injection volume 5 μL, and column temperature 30 °C. MS was performed in the conditions with a capillary voltage of 0.3 kV, a cone voltage of 25 V, the desolation temperature of 600 °C, a desolation gas flow at 950 L/h, a cone gas flow at 150 L/h, and a nebulizer at the pressure of 7.0 bar. The MS parameters of each compound are indicated in Table 2.

Determination of LOD: Blank samples (CAP-free tissues) were prepared, extracted, purified, and tested by LC-MS according to the same method mentioned above. In addition, standards in solvent were prepared by diluting CAP-D_5_ and CAP standard solutions with methanol, and tested by LC-MS according to the same method mentioned above. The calibration curve was plotted using the ratio of CAP to CAP-D_5_ response value (ordinate) against the concentration of standard solutions (abscissa). The calibration curve is shown in Figure 1. Finally, the ratio of signal to noise was obtained by combining the measured values of the blank sample and that for the standard curve, 3 times of which made the LOD.

### 2.3. Quality Control

Determined by a signal-to-noise ratio of 3:1, the limit of detection (LOD) in this study was 0.1 μg/kg. The limit of quantification (LOQ) was determined as the point where the signal-to-noise ratio was greater than 10:1. The recovery rate in each food class was established by the recovery of spiked CAP, which ranged from 80 to 110%.

### 2.4. Estimation of Daily Food Consumption

Food consumption data were obtained from a national food consumption survey of urban and rural residents of Guangzhou conducted in 2011. Information on dietary intake was based on a consecutive 3-day/24 h recall questionnaire in combination with the weighing method for edible cooking oil. Details of the methodology are available in our previously published manuscripts [13,29]. Briefly, 2960 residents from 998 households were surveyed in the study, among which 1416 were male and 1544 were female. Urban residents accounted for 63.8% of the total, and suburban residents represented 36.2% of the total study population. The ages of participants ranged from 3 to 86 years, with the mean age as 32 years. The age groups of 3 to 6, 7 to 17, 18 to 59, and 60 years and above accounted for 6.7, 21.5, 58.6, and 13.2% of the total study population, respectively [30,31,32].

### 2.5. Estimation of Daily Intake of CAP

The total dietary CAP intake was calculated as an estimated daily intake (EDI) by using Equation (1) [33].
(1)$$EDI=\overset{n}{\underset{i=1}{\sum }}\frac{D_{i}\times M_{i}}{W}$$

*EDI* is the estimation of daily intake of dietary CAP (mg/kg body weight per day). *D_i_* is the daily intake of each food in each age group (g/person∙day^−1^). *M_i_* is the mean level intake of CAP in each food category (μg/kg). When CAP was not detected in certain types of food, *M_i_* was then assumed to be LOD/2 [34]. *W* is the body weight of each respondent (kg). Twenty kilograms was determined to be the average weight of respondents at 3 to 6 years [35], 40 kg for respondents at 7 to 17 years [36], and 60 kg [22] for participants of the other two groups, 18 to 59 years and ≥60 years. The mean daily exposure to CAP was estimated by using @RISK software (from Palisade Coporation, Ithaca, NY, USA).

### 2.6. Risk Characterization

CAP is a genotoxic carcinogen. For genotoxic carcinogens that do not have thresholds, use of the MOE analysis technology for a risk assessment was recommended by the current Codex Alimentarius Commission (CAC), as well as the European Food Safety Agency [37]. Therefore, in this study we applied MOE analysis to assess the dietary exposure to CAP in Guangzhou citizens. The higher the MOE value, the lower the relevant risk to humans, and vice versa.

The MOE value of a genotoxic carcinogen is the ratio of the BMDL (Bench Mark Dose Lower Confidence Limit), which is obtained according to the 5% confidence interval of the benchmark dose (Bench Mark Dose, BMD), to the estimated human exposure [37]. For the reason that experimental data for CAP exposure in humans are rare, this study quoted the lower limit of the baseline dose proposed by the Global Aquaculture Alliance (GAA) on CAP residues in aquatic products, and the BMDL value was estimated to be 1 μg/kg per day for animal tumors with the most conservative method, so the BMDL value of CAP was 1 μg/kg per day in this study. Taking into account the weight difference between adults and children, 20, 40, 62, and 60 kg were used to calculate the exposure of CAP per kilogram of body weight and MOE values in the various age groups. MOE values < 5000, 5000~500,000, and >500,000 indicate high, moderate, and low levels of health risk, according to Health Canada’s MOE evaluation guidelines for genotoxic carcinogens [38].

### 2.7. Statistical Analysis

Mean ± S.D. was used to express the concentrations of CAP in foodstuffs, on which a descriptive statistical analysis was performed. Probabilistic risk assessment model calculations for CAP dietary exposure and MOE values were performed by using @RISK software (Palisade Corporation, 7.6. Industrial, 2018) based on a Monte Carlo simulation with 10,000 iterations. The results are displayed as the mean values and the ranges from the 5th to the 95th percentile.

## 3. Results and Discussion

### 3.1. Levels of CAP in Various Types of Foods

A total of 1454 samples of raw poultry meat, offals, fish, crustaceans, molluscs, egg and egg products, and liquefied milk were collected in Guangzhou from 2016 to 2019. As indicated in Table 3, CAP was detected in 248 of them (detection rate being 17.1%), with an average value of 29.1 μg/kg for the CAP-detectable samples. Out of all food categories tested, molluscs showed the highest average value, 148.2 μg/kg, with the top value being 8196 μg/kg. The lowest average value was observed in poultry samples (0.36 μg/kg), the top value being only 43.4 μg/kg. In recent years, CAP has been detected more and more frequently in live shellfish from various markets in China, some samples even contained CAP exceeding the Chinese national safety limit for foods [10,39]. CAP has been detected in several main types of shellfish, which are popularly consumed by citizens, including mussels, scallops, holy seeds, oysters, arctic shellfish, and sixties clam. Notably, the level of CAP in clam was particularly high, much higher than in other shellfish subcategories [40]. The high level of CAP in clam may not be primarily attributed to environmental pollution, but to artificial addition, since most clams sold in the Chinese markets are artificially cultivated at the seaside along coastal areas, such as Guangdong, Guangxi, Liaoning, Shandong, and Fujian. Clams grow rapidly and have a short breeding cycle and a long survival time out of water; hence, it is an excellent type of shellfish suitable for artificial, high-density farming and has become one of the four largest types of shellfish farmed as a food supply in China [41,42]. According to investigations, clams can survive for one or two days in humid environments outside water, and in order to prolong the survival time of the clam, wholesalers spray CAP and other antibiotic portions on clams, which has caused the level of CAP in clams to exceed the safety limit [1].

In a report, a comparison between the contamination of CAP in foods sold in Guangzhou with that in another city, Shanghai, which is located directly by the seashore, was made. The detection rate of CAP in aquatic products (fish, crustaceans, and molluscs) in Guangzhou was 24.65%, while that in Shanghai was only 0.34%, much lower than in Guangzhou [1]. Shanghai is located on the west coast of the Pacific Ocean and the east coast of the Asian continent. Both Guangzhou and Shanghai belong to China’s first-tier cities, with consumer megacities, rapid economic growth, rapid social development, large food circulation, low self-sufficiency, and high dependence on external sources, thus increasing food safety uncertainties. The contamination of poultry meat with CAP in Guangzhou with a detection rate of 9.0% appears also higher than in Shanghai, where the detection rate was 4.49%. On the contrary, the averaged value of CAP in poultry meat sold in Guangzhou and Shanghai was 0.36 and 1.28 μg kg^−1^, respectively, indicating a relatively high burden of CAP in the poultry meat sold in Shanghai [43]. Likewise, although the rate of detection of CAP in egg products sold in Guangzhou (1.43%) was quite similar to that in Shanghai (1.71%), the average value of 32.10 μg kg^−1^ in Shanghai was much higher than that of 0.05 μg kg^−1^ in Guangzhou. It appears that the distribution of CAP values in foods sold in the two cities might be different: in Guangzhou most values were relatively low, while in Shanghai many values were relatively high despite that most detections were negative [43].

In the recent three decades, with the rapid development of China’s economy, Chinese people’s living conditions have been greatly changed, with improved dietary structure and increased consumption of animal-based foods. There might be various factors responsible for CAP residues in animal foods, but the major one should be illegal addition, followed by environmental pollution. Due to the general characteristics of intensive farming in China, the high density of animal husbandry makes it easy for animals to spread diseases [44]. Although CAP is a banned veterinary drug in China, it is still used in the process of livestock breeding due to its low price, quick effect, easy availability, and the difficulty in identifying illegal CAP users and relatively “light” punishment given to them [15]. In addition, some breeders lack law-abiding awareness and focus only on economic benefits, thus some of them may not comply with the rule of a drug withdrawal period before slaughter [44]. Moreover, because the graze and drinking water (used for animal husbandry) might be contaminated with industrial pollutants and pesticides, they may contribute, to some (but probably insignificant) extent, to CAP burdens in domestic animals [3].

### 3.2. Dietary Exposure

Table 4 below lists the EDI of CAP from various foods in citizens at different ages in Guangzhou. In all age groups, CAP intake from pond fish consumption contributed to daily intake of CAP most heavily, depending on fish consumption amount by a typical citizen and the level of CAP it contains. Thus, pond fish was the primary source of dietary CAP exposure for residents in Guangzhou. Although pork had the largest intake (93.1 (25.7~161) g/day), but with low CAP content, its EDI was the second largest contributor to CAP exposure in Guangzhou residents. Poultry meat was the third leading cause of CAP exposure, while the EDI of molluscs was the lowest (with a high CAP level but extremely low consumption). Similarly, due to the low level of consumption, viscera, shrimp, and crab had quite small values of EDI.

The estimated EDI of CAP for citizens at varying ages ranged from 0.27 ng/kg b.w./day to 0.48 ng/kg b.w./day, and the average EDI was 0.4 ng/kg b.w./day (with a 90% confidence interval from 0 to 2.0 ng/kg b.w./day). Among the different age groups, the 3–6 age group had the highest EDI, and residents at 60 years and above had the lowest EDI. The main source of CAP exposure in all age groups was pork, but along the progression of age, the EDI showed a slowly decreasing trend.

In this study, the exposure of Guangzhou residents to CAP via consumption of molluscs, a minor dietary source of CAP, was only 0.005 ng/kg b.w./day. This is much lower than 9.65 ng/kg b.w./day, a value of EDI of CAP obtained by another report, where CAP intake from the consumption of commercial shellfish products in Guangzhou residents was evaluated [39]. However, the latter evaluation was based on consumption data indirectly deducted from other literature, rather than primarily investigating the target population, as done in the present study. Therefore, our evaluation on dietary CAP exposure in Guangzhou citizens is probably more accurate and reliable.

### 3.3. Estimated Health Risk of CAP Exposure in Guangzhou Citizens at Various Ages Indicated by MOE Value

Table 5 lists the exposure and relevant risk of CAP, indicated by EDI and MOE values, in Guangzhou citizens at various ages. The results indicated that the MOE values in all age groups were all less than 5000, and by probabilistic risk analysis the lower limit of each MOE value was 0, which suggests that the dietary exposure of Guangzhou citizens to CAP was unneglectable. This is worthy of close attention from the local food and environment administration.

Our results also indicated that children in the 3–6 age group had the lowest MOE value, which implies even greater risk in preschool children than the general population, at least in Guangzhou.

### 3.4. Uncertainty Analysis and Limitations of This Study

Ambiguity, to some extent, was unavoidably introduced in the process of the present food safety risk assessment. Firstly, with the uncertainty of exposure, the CAP content was directly measured from the samples purchased from each market, without considering potential changes in the CAP content of the food during its processing and cooking. In addition, although the study considered a variety of foods, CAP may also be present in other food commodities, such as honey, so the inability to accurately assess the actual exposure of CAP may have caused an underestimation of the real exposure levels. Secondly, there might be an uncertainty of the baseline dose; since the international community has not yet released relevant thresholds for CAP in human trials, in this study the lower limit of the BMD of CAP in aquatic products proposed by the International Fisheries Association (required for induction of animal tumors), i.e., 1000 ng/kg B.W./day, was adopted. This BMD value is only based on animal experiments, and there is still some uncertainty in the extrapolation from results of animal experiments to humans.

## 4. Conclusions

In summary, based on the risk assessment in this study, the dietary exposure of Guangzhou residents at various ages to CAP has a potential, noticeable health threat, particularly significant for children 3 to 6 years old. Consumption of pond fish, pork, and poultry meat contributes to the majority of dietary CAP exposure.

## Figures and Tables

**Figure 1 ijerph-18-08805-f001:**
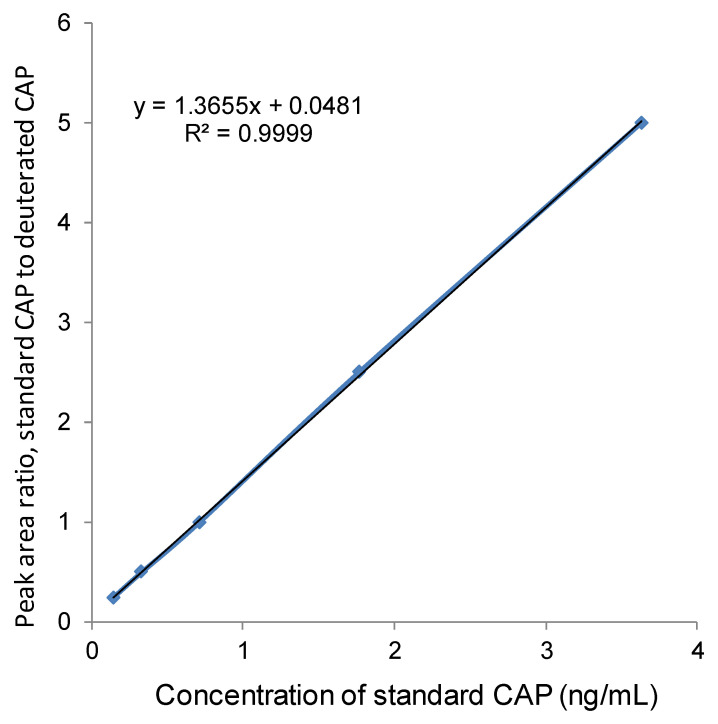
Calibration curve of CAP determination by LC-MS.

**Table 1 ijerph-18-08805-t001:** The procedure for gradient elution.

Time, min	0.05% Ammonia	Acetonitrile %
0	10	90
0.2	10	90
1.5	90	10
2.2	90	10
2.5	10	90
6.0	10	90

**Table 2 ijerph-18-08805-t002:** The parameters for analysis of each compound by MS.

Compound	Parent (m/z)	Daughter (m/z)	Cone (V)	Collision (V)
CAP	321.1	151.9	24	18
CAP	321.1	257.0	24	10
CAP-D_5_	326.1	157.1	30	18
CAP-D_5_	326.1	262.1	30	12

**Table 3 ijerph-18-08805-t003:** Levels of CAP in foods collected in Guangzhou from 2016 to 2019.

Food Category	Number of Samples	<LOD	Detection Rate (%)	Level of Chloramphenicol (μg kg^−1^)			
				Mean ± Standard Deviation	*P* _50_	*P* _95_	Range
Pork	74	74	0	0.05	ND	ND	ND
Beef/lamb	86	73	15.1	0.2 ± 1.4	ND	0.22	ND~12.80
Poultry	202	182	9.9	0.4 ± 3.2	ND	0.32	ND~43.40
Viscera	78	76	2.6	0.1 ± 0.1	ND	ND	ND~1.30
Pond fish	398	337	15.3	0.2 ± 0.5	ND	0.72	ND~4.92
Shrimp/crab	170	161	5.3	0.1 ± 0.3	ND	0.18	ND~2.53
mollsuk	288	147	49.0	148 ± 760	ND	780.70	ND~8196
Eggs	140	138	1.4	0.1 ± 0.03	ND	ND	ND~0.20
milk	18	18	0	0.05	ND	ND	ND
Total	1454	1206	17.06	29 ± 343	ND	1.78	ND~8196

ND: Not detected; LOD: Limit of detection; *P*_50_: the 50th percentile; *P*_95_: the 95th percentile.

**Table 4 ijerph-18-08805-t004:** EDI of CAP from food in Guangzhou citizens at different ages.

Food Category	DCRP (g/day) in Groups at Varying Age (Years)					EDI of CAP (ng/kg/Day) in Groups at Varying Age (Years)				
	3~6	7~17	18~59	≥60	Total	3~6	7~17	18~59	≥60	Total
Poultry	27.6 (0~59.3)	44.7 (4.1~85.3)	49.9 (1.9~97.9)	42.2 (5.1~79.3)	46.6 (1.3~91.9)	0.10 (0~0.4)	0.08 (0~0.3)	0.06 (0~0.2)	0.05 (0~0.2)	0.06 (0~0.2)
Pork	60.0 (16.8~103)	88.4 (24.0~153)	98.6 (28.7~169)	94.7 (33.2~156)	93.1 (25.7~161)	0.16 (0~0.5)	0.12 (0~0.4)	0.09 (0~0.3)	0.08 (0~0.3)	0.08 (0~0.3)
Beef/lamb	7.6 (0~23.0)	13.6 (0~36.6)	16.0 (0~42.7)	10.1 (0~27.5)	14.5 (0~39.4)	0.03 (0~0.14)	0.02 (0~0.16)	0.02 (0~0.09)	0.01 (0~0.05)	0.02 (0~0.06)
Viscera	2.7 (0~11.7)	4.8 (0~18.6)	5.7 (0~22.4)	6.6 (0~24.0)	5.3 (0~21.1)	0.007 (0~0.06)	0.006 (0~0.05)	0.005 (0~0.04)	0.006 (0~0.04)	0.005 (0~0.03)
Eggs	35.7 (4.5~66.9)	33.4 (0~66.9)	33.0 (0.9~65.1)	32.0 (4.6~59.4)	33.3 (1.3~65.3)	0.09 (0~0.24)	0.04 (0~0.12)	0.03 (0~0.08)	0.03 (0~0.07)	0.03 (0~0.08)
Milk	98.7 (0~210)	85.7 (0~188)	53.8 (0~133)	44.3 (0~109)	63.5 (0~152)	0.11 (0~0.33)	0.05 (0~0.15)	0.02 (0~0.07)	0.02 (0~0.06)	0.02 (0~0.08)
Pond fish	27.0 (0~56.7)	38.3 (2.4~74.2)	46.1 (3.2~89.0)	49.8 (3.3~96.3)	43.1 (1.9~84.3)	0.11 (0~0.4)	0.08 (0~0.3)	0.06 (0~0.2)	0.07 (0~0.2)	0.18 (0~0.6)
Mollsuk	0.8 (0~5.0)	2.0 (0~9.8)	2.2 (0~11.1)	2.3 (0~10.3)	2.1 (0~10.5)	0.006 (0~1.0)	0.007 (0~1.0)	0.005 (0~1.0)	0.005 (0~1.0)	0.005 (0~1.0)
Shrimp/crab	3.0 (0~11.6)	4.9 (0~18.5)	6.3 (0~26.7)	7.0 (0~24.8)	5.8 (0~24.2)	0.009 (0~0.05)	0.007 (0~ 0.04)	0.006 (0~0.04)	0.007 (0~0.03)	0.006 (0~0.04)
Total						0.48 (0~2.0)	0.41 (0~2.0)	0.29 (0~1.0)	0.27 (0~1.0)	0.40 (0~2.0)

EDI: Estimated daily intake, DCRP: dietary consumption per reference person. Data of the EDI of CAP are the 90% confidence interval. The content of CAP in each kind of food was determined using samples collected from 2016 to 2019, and the food consumption data were obtained in a study conducted in 2011 [13,29]. The lower limits of all the intervals were negative, which are supposed to be irrational, so they are expressed as zero.

**Table 5 ijerph-18-08805-t005:** Risk characterization of CAP exposure in Guangzhou citizens at different ages by MOE analysis.

Age (Years)	EDI of Chloramphenicol (ng/kg/day)	Margin of Exposure
3~6	0.5 (0~2.0)	2094 (0~8620)
7~17	0.4 (0~2.0)	2423 (0~9120)
18~59	0.3 (0~1.0)	3486 (0~13,243)
≥60	0.3 (0~1.0)	3703 (0~13,171)
Total	0.4 (0~2.0)	2489 (0~10,512)

Each MOE value was obtained by dividing BMDL (1000 ng/kg BW day) with EDI (estimated daily intake), of which the 90% of confidence interval was provided (in parentheses).

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
