# Peer review of "Probabilistic Risk Assessment of Dietary Exposure to Chloramphenicol in Guangzhou, China"

_ijerph, 2021, doi:10.3390/ijerph18168805_

Round 1

Reviewer 1 Report

I reviewed the manuscript “Probabilistic risk assessment of dietary exposure to chloramphenicol in Guangzhou, China” (ijerph-1348899). The manuscript is a revised version; however I suggest some minor changes:

a) The manuscript must have only one first author and one corresponding author. The other 12 researchers should be co-author. The manuscript is not interdisciplinary and then it is not justified the 4 first authors and 2 corresponding authors.

b) The authors modified EDI confidence intervals in Table 4. However, I suggest using the same criteria for DCRP values, there are some intervals below the zero, which is not possible. Please review carefully Table 4.

c) Please review carefully the references citation. There are still some mistakes, i.e. Reference 33

Author Response

Question #1:  The manuscript must have only one first author and one corresponding author. The other 12 researchers should be co-author. The manuscript is not interdisciplinary and then it is not justified the 4 first authors and 2 corresponding authors.

Answer: We understand and respect this reviewer's suggestion. However, we have important difficulties to make changes in the lists of first authors and corresponding authors. Reasons: 1) this work is based on the cooperation of two institutions, and there is a principal investigator in each institution, who is responsible for the study design, management of conduction, analysis of data, and manuscript writing; particularly, the size of samples is quite large, therefore, many experimenters are involved in the whole sample collection and analysis process, four of them made major and equal contribution to the sample collection/analysis and manuscript preparation. 2) Since the four fist authors and two corresponding authors had reached consensus on the  authorship, we feel impossible to make changes on this stage of manuscript submission.

Question #2:  The authors modified EDI confidence intervals in Table 4. However, I suggest using the same criteria for DCRP values, there are some intervals below the zero, which is not possible. Please review carefully Table 4.

Answer: This has been changed as suggested. Please see the new Table 4 in the revised manuscript.

Question #3: Please review carefully the references citation. There are still some mistakes, i.e. Reference 33.

Answer: We have checked through the references carefully, and have found no mistakes except for reference 33 (which was also noticed by this reviewer). Now we have changed the journal title into standard English expression.

Reviewer 2 Report

The authors have greatly improve their manuscript and I will recommend publication in its current format

Author Response

Question: The authors have greatly improve their manuscript and I will recommend publication in its current format.

Answer: Thanks to this reviewer, for her/his approvement on our efforts in making the manuscript better. 

Reviewer 3 Report

First of all, authors did not afford "points-by points" summaries for reviewers' comments for manuscript re-submission.  It doesn't make sense for peer-review systems.

1) Table 1and 2, and Figure 1 are not essential in the text body. These are summarized in supplemental data.

2) Details (include sample numbers) of each year data  for Table 3 must be listed in supplemental data. (especially mollusk data)

3) The data in Table 4, year range for data collection has to describe and the details of each year data  must be listed in supplemental data.

4) It doesn't make sense for "Risk characterization of CAP exposure in Guangzhou citizens" in table 5.

Did government have to establish the EDI and MOE for each city in China?

Author Response

Question #1: First of all, authors did not afford "points-by points" summaries for reviewers' comments for manuscript re-submission.  It doesn't make sense for peer-review systems.

Answer: This is not true, and inconsistent with the other two reviewers's comments. Clearly we have provided point-to-point responses to each question raised by each reviewer.

Question #2: Table 1and 2, and Figure 1 are not essential in the text body. These are summarized in supplemental data.

Answer: Tables 1 and 2, and Figure 1 were created according to reviewers' suggestions on the first run of review; they are critical supports on the methodology of our determination of chloramphenicol. Therefore, we would not move them to the supplemental materials.

Question #3: Details (include sample numbers) of each year data  for Table 3 must be listed in supplemental data. (especially mollusk data).

Answer: Since different types of foods were sampled in each year, it is not appropriate to make comparisons between different years, which would only make things odd and complicated.

Question #4: The data in Table 4, year range for data collection has to describe and the details of each year data  must be listed in supplemental data.

Answer: As suggested, we have added the range of years for sample collection in the legend of Table 4; besides, we also added the year and literature sources for the food consumption data representing Guangzhou citizens. As for the detailed data of each year, this has been answered in our response to question #3.

Question #5:  It doesn't make sense for "Risk characterization of CAP exposure in Guangzhou citizens" in table 5.

Answer: We think Table 5 is the center of risk analysis, which is the major object of this work. Therefore, we do not agree with this comment, and we do not want to remove Table 5 from the manuscript.

Question #6: Did government have to establish the EDI and MOE for each city in China?

Answer: EDI and MOE are actually parameters for food consumption and substance exposure estimate as reference to an established risk, respectively. They are not standards established by a goverment.

This manuscript is a resubmission of an earlier submission. The following is a list of the peer review reports and author responses from that submission.

Round 1

Reviewer 1 Report

I review the manuscript “Probabilistic risk assessment of dietary exposure to chloramphenicol in Guangzhou, China”. The authors include a statistical study of the results of analysis of an adequate number of samples in Guangzhou. However, the explanations of the results seem to be contradictory because of the redaction. I suggest using a professional English editor, there are several mistakes and the manuscript is hard to understand.

The article has: 14 authors, 4”first authors” and 2 corresponding authors. In my opinion, this is excessive for a single discipline manuscript which uses an statistical software for analysis of experimental data.

Please review carefully the analytical section, there are some strange experimental protocols which could affect the accuracy of the experimental data employed for the statistical analysis.

Some critical points.

Abstracti

Please do not use abbreviation in the abstract.

The definition of

Page 1. The authors use CAP (CAP) in the definition of chloramphenicol.

Section 2.2 is incomplete, the LC-MS conditions are not mentioned and the sample preparation must be carefully reviewed.

There is not a derivatization process in the methodology proposed. Please check and modify the text.

Pre-detection technique is a Solid phase extraction (SPE), which is part of the sample treatment. In this section, the solvent employed is the same for activation-retention-elution. Please add a clear explanation regarding the protocol followed. This is not congruent with SPE methodologies.  

Section 2.3. Please indicate the protocol employed for determination of LOD. It is required the addition of precision values. The LOD depends on the analytical matrix, please include the LOD values for each type of sample evaluated in spiked samples. This point is critical for accuracy of the data obtained.

Section 3.2, the redaction is not clear. MOE values are < 5000, please include an explanation why there is a high health risk. According to the previous idea, it seems the contrary.

In Table 2, there are DCRP data with high confidence intervals which includes sometimes zero. Please add information about an interpretation of negative values for DCRP.

Please double check the references, there are 30 references and then it begins again with number 1.

Many of the references must be revised, they are local Chinese journals. Please do a proper revision in indexed scientific journals, this work can be easily done between 14 authors of the manuscript.  

Reviewer 2 Report

The manuscript by Zhang et al., titled '' Probabilistic risk assessment of dietary exposure to chloramphenicol in Guangzhou, China''analyzed residue chloramphenicol levels in different food sources in Guangzhou city and determined the estimated daily intake of the residue drug in different age groups, wherein they found EDI to be high in 3-6 years old kids and high residual drug levels in molluscs.

In general, I find the work interesting, well written and easy to understand.

I have just few minor comments that need attention (see reviewed manuscript)

''its residues remain widely detectable in various food commodities, such as eggs, meat, milk and honey.'' This sentence needs a reference

 ''Due to its ubiquitous nature, many reports have indicated the persistence of residues of CAP as an organic pollutant in several relevant food products in China and globally''. This sentence needs a reference.

Which organism is called sixties?

What is crustaceans of livestock?

Reviewer 3 Report

First of all, reviewer can not access their supplementary data, so reviewer can not provide correct judgement for their submission.

But the text body will be reconstructed of significant figures of each values between text body and tables.

The revised manuscript will be contained each year trends for the residual CAP, not summary from 2016-2019.

All table is not easy to recognized because the each line and raw are complicated.

The reference styles must to reconstruct essentially. for example ref 24 : Environmental science and pollution research international. 2012 Mar;19(3):609-18. PubMed PMID: 21881906. Epub 2011/09/02. eng.
